# Distinct Mechanisms of Cytotoxicity in Novel Nitrogenous Heterocycles: Future Directions for a New Anti-Cancer Agent

**DOI:** 10.3390/molecules27082409

**Published:** 2022-04-08

**Authors:** Rasha Saad Suliman, Sahar Saleh Alghamdi, Rizwan Ali, Ishrat Rahman, Tariq Alqahtani, Ibrahim K. Frah, Dimah A. Aljatli, Sarah Huwaizi, Shatha Algheribe, Zeyad Alehaideb, Imadul Islam

**Affiliations:** 1College of Pharmacy, King Saud bin Abdulaziz University for Health Sciences, Riyadh 14611, Saudi Arabia; aliri@ngha.med.sa (R.A.); qahtanita@ksau-hs.edu.sa (T.A.); ibrahimfarh42@gmail.com (I.K.F.); aljatli097@ksau-hs.edu.sa (D.A.A.); 2Medical Research Core Facility and Platforms, King Abdullah International Medical Research Center (KAIMRC), Ministry of National Guard Health Affairs, Riyadh 14811, Saudi Arabia; Huwaizisa@ngha.med.sa (S.H.); sh.algheribe@gmail.com (S.A.); zehaideb@gmail.com (Z.A.); islamim@ngha.med.sa (I.I.); 3Department of Basic Dental Sciences, College of Dentistry, Princess Nourah Bint Abdulrahman University, Riyadh 11671, Saudi Arabia; imrahman@pnu.edu.sa

**Keywords:** Imidazole, oxazolone, ADME, target prediction, anti-cancer, tubulin inhibitors

## Abstract

Electron-rich, nitrogenous heteroaromatic compounds interact more with biological/cellular components than their non-nitrogenous counterparts. The strong intermolecular interactions with proteins, enzymes, and receptors confer significant biological and therapeutic properties to the imidazole derivatives, giving rise to a well-known and extensively used range of therapeutic drugs used for infections, inflammation, and cancer, to name a few. The current study investigates the anti-cancer properties of fourteen previously synthesized nitrogenous heterocycles, derivatives of imidazole and oxazolone, on a panel of cancer cell lines and, in addition, predicts the molecular interactions, pharmacokinetic and safety profiles of these compounds. Method: The MTT and CellTiter-Glo^®^ assays were used to screen the imidazole and oxazolone derivatives on six cancer cell lines: HL60, MDA-MB-321, KAIMRC1, KMIRC2, MCF-10A, and HCT8. Subsequently, in vitro tubulin staining and imaging were performed, and the level of apoptosis was measured using the Promega ApoTox-Glo^®^ triplex assay. Furthermore, several computational tools were utilized to investigate the pharmacokinetics and safety profile, including PASS Online, SEA Search, the QikProp tool, SwissADME, ProTox-II, and an in silico molecular docking study on tubulin to identify the critical molecular interactions. Results: In vitro analysis identified compounds 8 and 9 to possess the most significant potent cytotoxic activity on the HL60 and MDA-MB-231 cell lines, supported by PASS Online anti-cancer predictions with pa scores of 0.413 and 0.434, respectively. In addition, compound 9 induced caspase 3/7 dependent-apoptosis and interfered with tubulin polymerization in the MDA-MB-231 cell line, consistent with in silico docking results, identifying binding similarity to the native ligand colchicine. All the derivatives, including compounds 8 and 9, had acceptable pharmacokinetics; however, the safety profile was suboptimal for all the tested derivates except compound 4. Conclusion: The imidazole derivative compound 9 is a promising anti-cancer agent that switches on caspase-dependent apoptotic cell death and modulates microtubule function. Therefore, it could be a lead compound for further drug optimization and development.

## 1. Introduction

N-heterocyclic aromatic molecules [1] exist as a five-membered ring containing three carbon and two nitrogen atoms (imidazolone) or nitrogen and oxygen in a five-membered ring (oxazolone). Previous studies of imidazolone and oxazolone [2], with their partially or fully saturated derivatives, highlighted such compounds as highly useful heterocycles [3], with applications in the chemical, biological, material, and pharmaceutical industries [4]. Imidazolone and oxazolone are an important class of heterocycles that are broad in their variety and application, and most are substances of interest due to their chemical and biological properties [5]. They are part of many highly significant biomolecules, such as the essential amino acid histidine and related compounds, biotin, and imidazole alkaloids [5].

In medicinal chemistry and as part of the drug delivery process, strategies to make active synthetic compounds are continually being developed. One particularly well-known method is to insert an imidazolone or oxazolone nucleus [6]. Imidazolone-based drugs have broad applications in many areas of clinical medicine [7]. One of the examples is cancer, which is the predominant driver of mortality rates, with the annual number of new cases estimated to reach 27.5 million annually by 2040 [7]. Presently, most current treatments include the combination of a chemotherapeutic agent, radiotherapy, surgery, or hormonal therapy [8]. Several chemotherapeutic agents possess imidazolone or oxazolone nuclei, such as the drug methotrexate (Figure 1), a well-known and therapeutic imidazole derivative acting as an inhibitor of DNA synthesis by preventing folate metabolism [9]. Methotrexate is clinically used to treat several neoplasms, such as acute lymphoblastic leukemia, acute myeloid leukemia, non-Hodgkin’s lymphoma, breast cancer, and bladder cancer [10,11]. Imidazolone is a highly polar and amphoteric species, having acid and base-like properties, causing it to be readily soluble in an aqueous environment, and the fact that it possesses an electron-rich core means it can easily accept and donate electrons, allowing strong biochemical interactions with a broad range of targets within a cell with greater affinity [12]. In addition, imidazoles have some structural similarities to purine and pyrimidine bases and other naturally occurring biologically active molecules. Such similarities allow these compounds to be used pharmacologically as active anti-metabolites and cytotoxic drugs, targeting viral, bacterial, and fungal infections and cancer [12,13].

Indeed, adding or conjugating imidazoles to other compounds is common to increase solubility, bioavailability, and interaction sites [14] and for selective targeting [15]. In addition, imidazolone derivatives are used to produce coordination complexes, as in platinum, where cisplatin is a therapeutic cancer treatment that acts similarly to the alkylating agents [16]. One particular study showed that a chalcone-imidazolone conjugate had cytotoxic properties, inducing apoptosis through DNA damage [17]. Moreover, oxazolone is primarily employed to treat antimicrobial-resistant Gram-positive bacterial infections caused by methicillin-resistant *staphylococci*, vancomycin-resistant *enterococci*, and penicillin-resistant *pneumococci* [18]. In addition, linezolid is also used to treat multidrug-resistant *Mycobacterium tuberculosis*. Jadomycin B^®^ (Figure 1) has cytotoxic and antibacterial properties, while deflazacort^®^ contains an oxazolone scaffold derived from prednisone, having anti-inflammatory and immunosuppressive effects [19]. The detailed biological activities of imidazole and oxazolone compounds have been reported previously [20,21].

Fourteen new imidazolone (Figure 2) and oxazolone derivatives were synthesized and chemically characterized without investigating their biological activities [22,23]. Therefore, this study focused on exploring the compounds’ in vitro and in silico anti-cancer activities to identify any lead candidate that may be further developed to provide an efficacious and potent selective anti-cancer drug therapy.

## 2. Results and Discussion

### 2.1. The Effect of Nitrogen-Based Heterocyclic Derivatives on Cytotoxicity

In vitro anti-proliferation assays are commonly used to evaluate the cytotoxicity of molecules against selected cell lines [24]. The MTT assay and CellTiter-Glo^®^ Assay (Promega™, Madison, WI, USA) were utilized as functional assays for evaluating the cytotoxic efficacy and potency (IC_50_) of test compounds against the selected cell lines [25]. The MTT assay was the first rapid colorimetric assay developed in the 1980s for high-throughput cell viability screening in a 96-well format and is still considered a gold standard [24,26]. However, the MTT assay has limitations, such as chemical interference, toxicity, and lack of sensitivity [27]. In addition, it is an endpoint assay that can only detect viable cells at a fixed time point. Consequently, the CellTiter-Glo^®^ assay was used to validate the anti-proliferative effects of the investigated compounds.

#### 2.1.1. Cell Viability and Proliferation Analysis

To establish the cytotoxic effect of the nitrogenous derivatives, cells were treated with various concentrations of the drugs ranging from 0 to 250 μM. Following a 24 h incubation period, cell viability was assessed using the MTT assay. Mitoxantrone served as a positive control to evaluate growth inhibition. Since heterogeneity is a hallmark of cancer, the study included cell lines with diverse genomic and proteomic profiles. MDA-MB-231(triple-negative breast cancer) demonstrated a great sensitivity towards compounds 8 and 9, with an IC_50_ of 4.7 μM and 17.02 μM, respectively (Figure 3a). Meanwhile, the non-malignant breast epithelial cell line (MCF-10A) showed reduced sensitivity towards compound 8 (Figure 3b). The IC_50_ was six-fold higher than that observed in the MDA-MB231 cells, indicating cytotoxic selectivity, a promising feature needed in any therapeutic anti-neoplastic agent. Compound 9, however, displayed a potent anti-proliferative effect in MCF-10A cells (IC_50_ = 9.516 μM). In this regard, further optimization would be required to increase the selective therapeutic potential of compound 9. To determine whether the anti-proliferative effect of the nitrogen derivatives could be translated to other malignancies, we tested the cytotoxic effect on the HCT8 (colorectal cancer) cell line. HCT8 is an adenocarcinoma driven by the Kirsten Rat Sarcoma (KRAS) mutation, unlike MCF-10A [28]. However, the cytotoxic effects were comparable to that observed in the MCF-10A (Figure 3c), indicating that the mechanism of the anti-proliferative effect of compounds 8 and 9 in HCT8 could be unrelated to the KRAS pathway. Data are summarized in Table 1.

#### 2.1.2. Cytotoxicity Evaluation of Nitrogen-based Derivatives using CellTiter-Glo Assay

To validate the anti-proliferative activity of the imidazole derivatives, the compounds were tested against several cancer cell lines, which included breast cancer (MDA-MB-231, KAIMRC1, and KAIMRC2), colorectal cancer (HCT8), and acute promyelocytic leukemia cells (HL60). KAIMRC1 was isolated from an Arab woman with stage II-B breast cancer and is ER+/PR+ and HER2-. In comparison, KAIMRC2 was isolated from a 34-year-old woman with metastasized breast cancer and is a triple-negative (ER-/PR-, and HER2-) tumor cell line [29,30].

Corroborating the MTT assay results, compounds 8 and 9 displayed a significant anti-proliferative effect after a 24 h treatment in MDA-MB-231 cells, with an IC_50_ of 18.97 μM and 39.19 μM, respectively (Figure 4). The KAIMRC1 cells demonstrated an increased sensitivity to compounds 8 and 9, with an IC_50_ of 10.20 μM and 22.18 μM, respectively. Meanwhile, KAIMRC2 also was sensitive to compound 9, with an IC_50_ of 27.11 μM. Both KAIMRC cell lines displayed differing proteomic phosphorylation statuses compared with MDA-MB-231, featuring constitutive activation of an Akt enzyme that is a protein kinase B (PKB), also known as (AKT), a type of serine/threonine-protein kinase, in a ligand-independent manner [29,30]. Phosphorylated AKT (pATK) plays a crucial role in cell survival and proliferation, cell cycles, and cellular invasion. It may be possible that nitrogen-based derivatives disrupt the AKT / mTOR pathway, which is overexpressed and activated in KAIMRC1 and KAIMRC2. Furthermore, the HL60 cells showed the greatest sensitivity to compounds 8 and 9, with an IC_50_ of 9.23 μM and 8.63 μM, respectively. These results are summarized in Table 2.

#### 2.1.3. High-Content Imaging (HCI)

We performed high content imaging on these cells since the HL60 cells were previously shown to exhibit the greatest sensitivity to compounds 8 and 9. Figure 5A depicts a dose-dependent increase in cell shrinkage/condensing, and necrotic cells are also visible. Treatment with increasing concentrations of compounds 8 or 9 led to apoptosis compared with control. Furthermore, even at the lowest concentration, 31.25 µM, compound 9 was more efficacious than compound 8 since there appeared to be fewer healthy cells (blue and green staining) and a greater level of shrinkage. Moreover, Figure 5B depicts the HCI Cell Health determined by means of digital analysis, displaying the HL60 percentage cell viability as a function of increasing drug concentrations. Together, these results identified both compounds 8 and 9 to be cytotoxic and to induce apoptosis; however, compound 9 was more efficacious and potent than compound 8.

#### 2.1.4. Effects of Compounds 8 and 9 on Microtubular Networks

An established cytotoxic mechanism of action of some nitrogenous heteroaromatic compounds is through the inhibition of tubulin. Normal microtubule function is essential during mitosis for mitotic spindle formation to allow chromatid separation, without which cell replication would cease. In addition, microtubules are critical for maintaining the cytoskeletal integrity of normal cells. Currently used cytotoxic anti-neoplastic classes of drugs such as the taxanes and vinca alkaloids also either inhibit microtubule assembly or disassembly, thus preventing mitosis from occurring. Therefore, we investigated the effects of compounds 8 and 9 on the modulation of microtubule function by assessing their effect on tubulin. As shown in Figure 6, compound 9 interfered with the microtubular network at the tested concentrations, 15.6 and 31 µM. Furthermore, at 31 µM, compound 9 exerted a nearly complete microtubule inhibition, comparable to the positive control, mitoxantrone. Surprisingly, compound 8 had no effect on tubulin at the concentrations tested; therefore, the cytotoxic mode of action of compound 8 was distinct from that of compound 9.

Several clinically utilized anti-cancer medications, such as paclitaxel, vinblastine, and vincristine, are imidazole derivatives and directly inhibit tubulin function [31,32]. Additionally, the colchicine-binding-site inhibitors (CBSI) are generally more flexible to modifications than other tubulin inhibitor types due to their structural simplicity [33]. A study investigated a panel of imidazole derivatives as tubulin polymerization inhibitors and identified that compounds targeting the colchicine-binding site were amongst the most efficacious cytotoxic agents, displaying dose-dependent effects causing cell cycle arrest at the G2/M phase [33,34]. Furthermore, the current study also demonstrated compound 9 to have cytotoxic effects via the inhibition of tubulin polymerization. Thus, we provided further supporting evidence from computational predictions using SEA Search and a tubulin docking study to identify the most probable binding mode of compound 9 to tubulin and compared it to that of the native ligand, colchicine.

MDA- MB 231 cells were treated with compounds 8 and 9 at two concentrations, 15.6 and 31.25 µM, for 48 h. The cells were stained with tubulin tracker™ Green (Cat #T34075) in HBSS for 30 min and nucleus stain HOECHST33342 (blue) for 5 min. The samples were imaged with Zeiss laser-scanning 780 microscopes.

#### 2.1.5. Apoptosis

The ApoTox-Glo^®^ triplex assay was utilized to further evaluate the effect of compound 9 on the MDA-MB-231 cancer cell viability, cytotoxicity, and apoptosis. This assay employs a luminogenic peptide substrate for caspase 3/7 to measure caspase activity, which is used as a marker for apoptosis within the cells. Treatment with compound 9 induced caspase 3-/7-mediated apoptosis in a dose-dependent manner in MDA-MB-231 cancer cell lines and inversely affected the viability of the cells, as expected (shown in Figure 7A). However, since this assay was time-dependent, the level of cytotoxicity (Figure 7B) determined, in this case, was not a true representative of the compound’s cytotoxic potential; it is reasonable to say that at the endpoint of apoptosis (Figure 7C), a heightened level of cytotoxicity would be observed.

One particular study evaluated the apoptosis induced by tri-substituted-imidazole in human breast cancer cells. The investigation probed the effect of 2-chloro-3-(4, 5-diphenyl-1H-imidazole-2-yl)-pyridine on procaspase-3 and found a substantial decline in the expression of cyclin D1, VEGF, survivin, and Bcl-2 proteins in a time-dependent manner [28,35], indicating a commitment to the apoptotic pathway. Likewise, imidazole’s induction of apoptosis in HL60 cells was associated with intracellular acidification, caspase-3 activation, and DFF-45 cleavage, which demonstrated the induction of caspase-dependent cell death [29,36], which is in agreement with the observations seen with compound 9 in our study.

### 2.2. Computational Studies

#### 2.2.1. Anti-Cancer Activity and Molecular Target Prediction

The PASS Online web server was used to predict the anti-cancer activity of the 14 compounds. As shown in Table 3, compounds 8–11, 13, and 14 demonstrated the highest probability as potential anti-cancer agents, while the rest showed low-to-inactive activity predictions. Moreover, the potential molecular targets for the synthesized compounds were predicted using the SEA Search. A higher similarity score (maxTC) with a lower significance score (*p*-value) indicates a higher probability that the protein is a potential target. This study focused on tubulin as a potential target since earlier reports identified imidazole-based derivatives as tubulin polymerization inhibitors [37,38,39,40]. Table 3 shows that compounds 8 and 9 possessed a high maxTC score (0.32 and 0.34, respectively) with a low *p*-value (1.544 × 10^−6^ and 1.137 × 10^−6^, respectively), indicating potential interference with microtubule-associated protein tau, which stabilizes the microtubule bundles and modulates the microtubule network assembly [41]. Compounds 11 and 13 were also predicted to have anti-cancer activity. However, the in vitro studies demonstrated that these compounds were far less potent than compounds 8 and 9.

#### 2.2.2. Molecular Docking and Dynamic Simulation with Tubulin Crystal Structure

Compounds 8 and 9 were selected for the docking study to investigate the molecular binding interaction with the tubulin crystal structure. The rationale behind our selection was to understand the influence of structural differences between compounds 8 and 9 that affected the results observed in the in vitro tubulin experiment. A methoxy group in compound 9 appeared to be the only source of the variance observed in these compounds’ inhibition of tubulin polymerization, where compound 9 successfully inhibited tubulin polymerization, while compound 8 did not. Thus, the Glide Schrodinger software was utilized to perform the molecular docking and binding free energy calculations for the docked complexes (results summarized in Table 4). Our results showed that both compounds occupied similar binding pockets to the native ligand (Colchicine) and positive control (mitoxantrone, Figure 8A); however, the interactions between compounds 8 and 9 were distinct. Compound 9 maintained hydrogen bond interactions with Cys 241 and VAL238 (via bridging water, Figure 8B), which were comparable to the native ligand and positive control. However, this was not the case for compound 8, which lacked these two significant interactions due to the absence of the methoxy group (Figure 8C). The docking results suggest that the presence of the methoxy group is critical for inhibiting tubulin polymerization, explaining why the interaction with tubulin was different for both compounds, although both demonstrated potent cytotoxic activities. Moreover, the binding free energy calculations for the docked poses demonstrated a comparable binding free energy score for compound 9 and mitoxantrone were −42.35 and −42.31, indicating a stronger binding with tubulin (Table 4) and confirming the above-mentioned experimental results.

To evaluate the stability of the molecular interactions, a molecular dynamic simulation run was performed for compound 9 in complex with the tubulin crystal structure (docked pose) for a timescale of 100 ns. The results identified that the ligand-protein complex was stable over the simulation run as there were no fluctuations in the RMSD values (Figure 9A). Moreover, the interaction contacts maintained for more than 50% of the simulation time were Cys 241, Gln 247, Leu 248, Leu 255, and Ala 354 (Figure 9B). In addition, the ligand interaction diagram showed that the ligand-amino acid contacts were consistent with our docking results, where the methoxy group in compound 9 maintained the interaction with Cys 241 and Val 238 via bridging water (Figure 9C), further confirming the importance of having a methoxy group for the inhibition of tubulin polymerization.

#### 2.2.3. Predictions of ADME Properties

Pharmacokinetic ADME properties are crucial for successful drug development and lead optimization. ADME parameters can influence the pharmacodynamics of a drug; for instance, the most common cause of low oral absorption is low solubility and permeability. Similarly, introducing hydrogen bond acceptors in the structure of a drug enhances the solubility and bioavailability of that compound [42,43]. Therefore, we utilized the SwissADME and Qikprop computational tools to make ADME predictions for the 14 compounds. As shown in Table 5, most compounds demonstrated high GI absorption, except for compounds 9–12 that possessed intermediate GI absorption (>80% is high, 80 to 25 is intermediate, and <25% is poor). The BBB penetration and CNS activity were predicted using a –2 (inactive) to +2 (active) QikProp scale. Compounds 8 to 12 demonstrated no CNS activity or BBB penetration, while the remaining compounds may potentially have intermediate effects. Additionally, all the compounds were inactive as P-glycoprotein (P-gp) substrates, suggesting none of the compounds would be effluxed out of the cancer cells upon treatment [44]. Moreover, the lipophilicity and solubility parameters for all the compounds were within the acceptable range required for orally bioavailable drugs (Log P is –2.0 to 6.5, and Log S is –6.5 to 0.5).

#### 2.2.4. Safety Profile Analysis—CYP P450 Enzyme Inhibition

Cytochrome P450 enzymes, including CYP1A2, CYP2C9, CYP2C19, CYP2D6, and CYP3A4, are the most predominant and crucial enzymes for metabolizing ninety percent of all known drugs [45]. Therefore, it is essential to assess and evaluate the effects of the 14 compounds on the CYP enzyme activity. The results summarized in Table 6 show that most of the compounds were predicted to inhibit CYP 1A2, 2C19, and 2C9, while only a few inhibited CYP 2D6 or CYP 3A4, including compound 9.

#### 2.2.5. Organ and Endpoint Toxicity Analysis

ProTox-II webserver provides the toxicological pathways and toxicity targets that predict the possible molecular mechanism behind the toxic response [46]. The lethal dose (LD_50_), organ toxicity, toxicity endpoints (carcinogenicity, mutagenicity, and immune-toxicity), Tox21 Nuclear receptor signaling pathways (Aromatase, Estrogen Receptor Alpha (ER), and Estrogen Receptor Ligand-Binding Domain (ER-LBD)), and Tox21 Stress response pathways (Mitochondrial Membrane Potential (MMP)) were predicted for the 14 synthesized compounds. As summarized in Table 7, the oral toxicity prediction findings showed that the majority of the compounds were categorized as class 4 (Harmful if swallowed (300 < LD_50_ ≤ 2000)), except for compounds 2, 4, 5, and 7, which were in class 5 (May be harmful if swallowed (2000 < LD_50_ ≤ 5000)) indicating a lower oral toxicity potential. Moreover, most of the compounds were potentially hepatotoxic, with probabilities ranging from 0.55 to 0.69 except compound 4, which was inactive. In contrast, all compounds were non-immunotoxic except compounds 9 and 10. Furthermore, compounds 8–13 were mutagenic with probability values ranging from 0.51 to 0.82, except for compound 14, which was classified as non-mutagenic. Among the compounds investigated, 13 out of the 14 compounds were predicted to be carcinogenic, with probability values ranging from 0.54 to 0.73. Compound 1 showed the highest toxic probability of 0.99 for Estrogen Receptor Alpha (ER) and 1 for both Aromatase and Estrogen Receptor Ligand-Binding Domain (ER-LBD). Thus, most of the compounds may be harmful when taken orally, including compounds 8 and 9, and they are all predicted to cause organ and endpoint toxicity except compound 4. Despite this, there is scope for further structural optimization to achieve safer oral compounds that cause minimal organ and endpoint toxicity.

## 3. Materials and Methods

### 3.1. Nitrogenous Heterocycle Samples

The fourteen (14) nitrogenous heterocycle samples were gifted by Prof. Dr. Ahmed Elsadig Mohammed Saeed from the college of chemistry, at Sudan University of Sci-ence and Technology. The detailed chemical synthesis of the samples together with their purity determination measures were conducted using TLC, UV-Vis, IR, and H^1^NMR. The complete filing of these information was explained in detail in the master thesis of Mr Ibrahim Khalifa Idriss Frah [23].

### 3.2. Anti-Cancer Activity Investigation

#### 3.2.1. MTT Assay

The MTT assay was utilized to examine the imidazole derivatives’ anti-proliferative activity against MCF-10A, MDA-MB-231, and HCT8 cell lines, purchased from ATCC, USA [47]. Firstly, 5 × 10^3^ cells/well were seeded in a 96-well plate in 100 μL of growth medium and were subsequently cultured overnight with various concentrations of each compound ranging from 0 to 250 µM [48]. The media was aspirated, and 50 µL of serum-free media and 50 µL of MTT solution were added to each well. Next, cells were incubated at 37 °C for 3 h, then 150 µL of MTT solvent was added into each well. After that, plates were wrapped in foil and agitated for 15 min on an orbital shaker. Finally, the absorbance was measured within 1 h at OD590 nm. Half-maximal inhibitory concentration IC_50_ values (μM) were calculated for each compound from the dose–response curve. Experiments were performed in triplicates.

#### 3.2.2. CellTiter-Glo Assay

The CellTiter-Glo assay (Promega™, Madison, WI, USA) was used to evaluate imidazole and oxazolone derivatives cytotoxicity against various cancer cell lines; MCF-10A, HL60, MDA-MB-231, and HCT8, purchased from ATCC, USA, except KAIMRC1 and KAIMRC2, which were isolated, established, and characterized in the core laboratory facility KAIMRC, Riyadh, Saudi Arabia. The cells were seeded in white 96-well plates at a density of 5 × 10^3^ cells/well in a 100 μL of growth medium with various concentrations of each compound ranging from 0 to 250 µM. Cells were incubated at 37 °C for 24 h. Plates were equilibrated at RT for 30 min. A quantity of 100 µL of CellTiter-Glo reagent was added to each well and mixed for 2 min on an orbital shaker. Plates were incubated for 10 min at RT before measuring the luminescence using an Envision plate reader (Perkin Elmer). Half-maximal inhibitory concentration IC_50_ values (μM) were calculated for each imidazole derivative from the dose–response curve [49]. The cytotoxic effect at a concentration <10 µM was considered strongly active and from 11 to 100 µM was considered moderately active, whereas above 100 µM was deemed non-active. Mitoxantrone was utilized as a positive control. Experiments were performed in triplicates [50].

### 3.3. High Content Imaging

HL60 cells were seeded in a 96-well plate at a density of 20,000 cells per well. The cells were treated with compounds 8 and 9 and negative control (PBS) for 48 h. Three graded concentrations were used for the treatment, 31.25 µM, 62.5 µM, and 125 µM. After the treatment, the cells were stained with Calcein AM (2 µg/mL), HOECHST33342 (2.5 µg/mL), and Propidium Iodide (2.5 µg/mL) for 45 min at 37 °C and 5% CO2. Plates were imaged on a Molecular Devices ImageXpress^®^ Microsystem, and the acquired image data were analyzed using MetaXpress^®^ software, Molecular Devices, Downingtown, PA, USA. The Cell Health module available in the MetaXpress software was used to measure the percentage viability of live and dead cells. Experiments were performed in triplicates.

### 3.4. Tubulin Staining and Imaging

Following the standard protocol, tubulin and microtubule-associated proteins (MAPs) were developed, accompanied by two cycles of polymerization and depolymerization [41]. MDA-MB231 cells were treated with compounds 8 and 9 at concentrations of 31.25 and 15.6 µM for 48 h. Subsequently, the media was replaced with HBSS, and the MDA-MB231 cells were stained with tubulin tracker™ Green 51(Cat #T34075) for 30 min and HOECHST33342 for 5 min. The Zeiss laser-scanning 780 microscope was used for imaging [51].

### 3.5. Apoptosis

The Promega ApoTox-Glo^®^ triplex assay was conducted as described by the manufacturer. Briefly, the cells were incubated at 37 °C for 12 h and then treated for 24 h with various concentrations of compound 9 [29,30]. Next, 100 µL of the Viability and Cytotoxicity Reagent was added to each well and briefly mixed on an orbital shaker. Plates were incubated at 37 °C for 30 min, and the fluorescence was measured at the following two-wavelength sets—400Ex/505Em (Viability) and 485Ex/520 Em (Cytotoxicity)—using the Envision plate reader (Perkin Elmer). Subsequently, the Caspase-Glo^®^ 3/7 Reagent (100 µL/well) was added, plates were briefly mixed on an orbital shaker (300 to 500 rpm for ~30 s), followed by an extra 30 min incubation at RT. Finally, to determine the level of apoptosis, the luminescence associated with caspase 3/7 activation was measured using an Envision plate reader (Perkin Elmer) [52,53].

### 3.6. Computational Methods:

#### 3.6.1. Anti-Cancer Activity Prediction

The anti-cancer activity of the synthesized compounds was predicted using the online webserver, prediction of activity spectra for substances (PASS), which utilizes a known actives database (http://way2drug.com/passonline/) (accessed on 22 November 2021) [54].

#### 3.6.2. Molecular Target Predictions

A similarity ensemble approach (SEA) search web server (https://sea.bkslab.org/) (accessed on 30 November 2021) was used to investigate the potential molecular targets for the synthesized compounds, focusing on tubulin as a potential target. The webserver utilizes a quantitative classification and target association according to the chemical similarity of protein-related ligands. It creates a list of Max Tanimoto coefficients (MaxTc), and E-values used to interpret results [55].

#### 3.6.3. Molecular Docking and MM-GBSA Binding Free-Energy Calculations with Tubulin Crystal Structure

The molecular interactions between compounds 8 and 9 and tubulin were investigated. Molecular docking was performed using the tubulin crystal structure (PDB: 4O2B), and Maestro Schroödinger software (Schrödinger Release 2021-4) as previously described [56]. The ligands were prepared using the LigPrep tool (LigPrep, Schrödinger, LLC, New York, NY, USA, 2021), and the protein was minimized and optimized using the Protein Preparation Wizard (Protein Preparation Wizard; Epik, Schrödinger, LLC, New York, NY, USA, 2021). After grid generation, the derivatives were docked into the colchicine-binding site, and post-docking analysis was performed for the docked compounds using the Glide tool (Glide, Schrödinger, LLC, New York, NY, USA, 2021). Moreover, the binding free energy calculations were conducted for the docked complexes using Prime MM-GBSA, VSGB as the solvation model, and an OPLS4 force field.

#### 3.6.4. Molecular Dynamic Simulation with Tubulin Crystal Structure

To investigate the stability of the docking interactions, we performed a molecular dynamic simulation run for the best docking pose for tubulin and compound 9. Desmond was utilized to run the simulation, and the TIP4P water molecules were added to the complex (21206 water molecules). The system was neutralized by adding 28 Na+, and the NPT ensemble was utilized to run the simulation. The complex was relaxed before the production run, and the timescale for the simulation run was 100 ns. Several simulation frames and simulation interaction diagrams (SID) were utilized for the analysis.

#### 3.6.5. Prediction of ADME/T Properties

Two computational tools were utilized to predict the absorption, distribution, metabolism, and excretion (ADME) of the synthesized compounds, SWISSADME (http://www.swissadme.ch/) (accessed on 14 November 2021) [57] and QikProp (QikProp, Schrödinger, LLC, New York, NY, 2021). The important pharmaceutical properties that were selected for evaluation were gastrointestinal (GI) absorption, blood–brain barrier (BBB) permeability, P-glycoprotein (P-gp) substrate, lipophilicity (log Po/w), and solubility (Log S).

#### 3.6.6. Safety Profile Analysis

##### CYP P450 Enzyme Inhibition

The SWISSADME server was used to conduct cytochrome (CYP) P450 enzyme-inhibition prediction for each compound against several CYP enzymes: CYP1A2, CYP2C19, CYP2C9, CYP2D6, and CYP3A4 57.

##### Organ and Endpoint Toxicity Analysis

The ProTox-II online tool toxicity prediction test was used to examine the safety profile for the synthesized compounds [46]. This server categorized compounds into six toxicity classes (1–6) with a prediction of the lethal dose (LD_50_) (mg/kg) and toxicity class based on available online databases. Class one possesses lethal toxicity with an estimated lethal dosage (LD_50_) of 5, and class six demonstrates an LD_50_ > 5000, indicating the compound is less toxic. Moreover, the webserver also determines each evaluated ligand’s organ and endpoint toxicity (https://tox-new.charite.de/protox_II/) (accessed on 3 November 2021).

## 4. Conclusions and Future Direction

The current study illustrates the in vitro anti-cancer activity of several nitrogenous heteroaromatic compound derivatives of imidazole and oxazolone. Compounds 8 and 9 provided potent cytotoxic activity in various cancer cell lines via distinct mechanisms. Compound 9 acted as a ligand for tubulin that was similar to colchicine, inhibiting microtubule function and subsequently preventing mitosis and inducing apoptosis through a caspase-dependent pathway, unlike compound 8. Molecular docking and simulation studies identified that the methoxy group in compound 9 was responsible for tubulin inhibition. Other computational studies helped identify compounds for lead-optimization based on ADME and safety profiles, including CYP enzyme inhibition and oral and organ toxicity. Our results revealed that compound 9 demonstrated an acceptable pharmacokinetic ADME profile based on Lipinski’s rule of five (ROF), although optimization would be necessary to improve this lead’s safety and ADME profile. Moreover, we believe post-safety and ADME optimization translation into an in vivo system would better reflect the therapeutic potential of this lead drug in a human biological system.

## Figures and Tables

**Figure 1 molecules-27-02409-f001:**
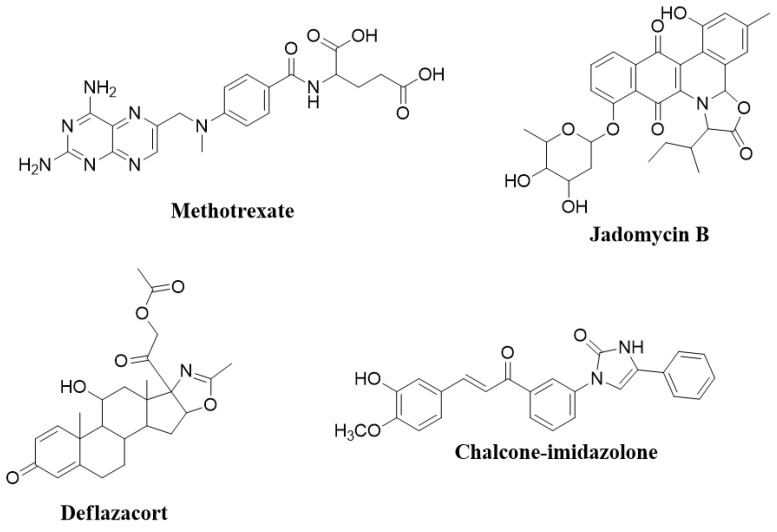
Bioactive molecules that contain imidazolone and oxazolone scaffolds.

**Figure 2 molecules-27-02409-f002:**
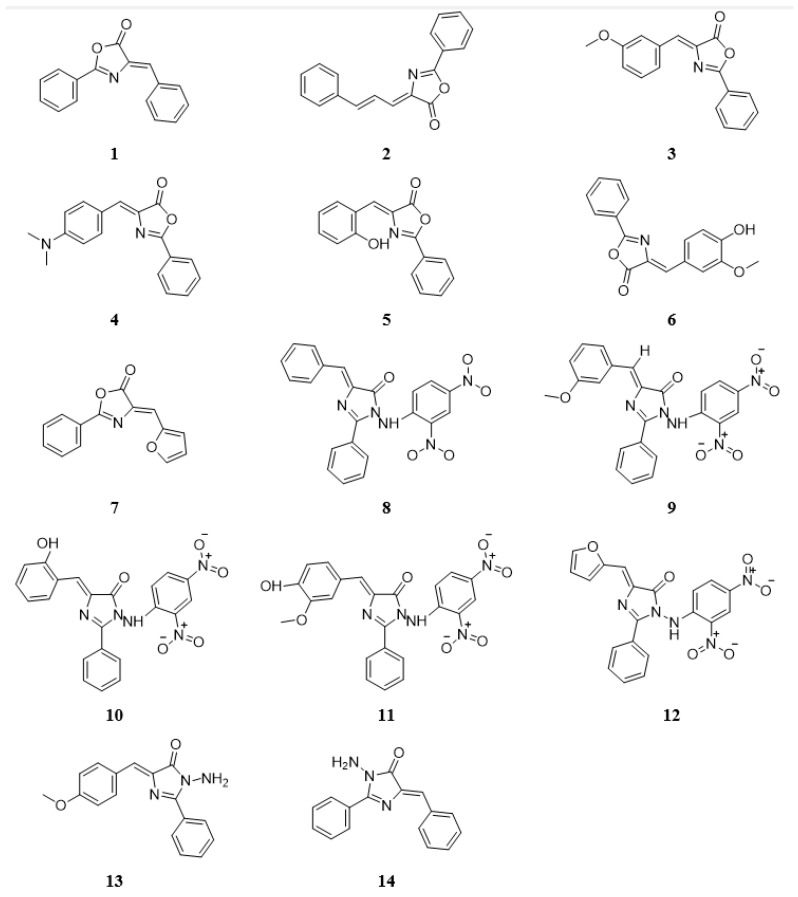
The chemical structures of nitrogen-based heterocyclic derivatives.

**Figure 3 molecules-27-02409-f003:**
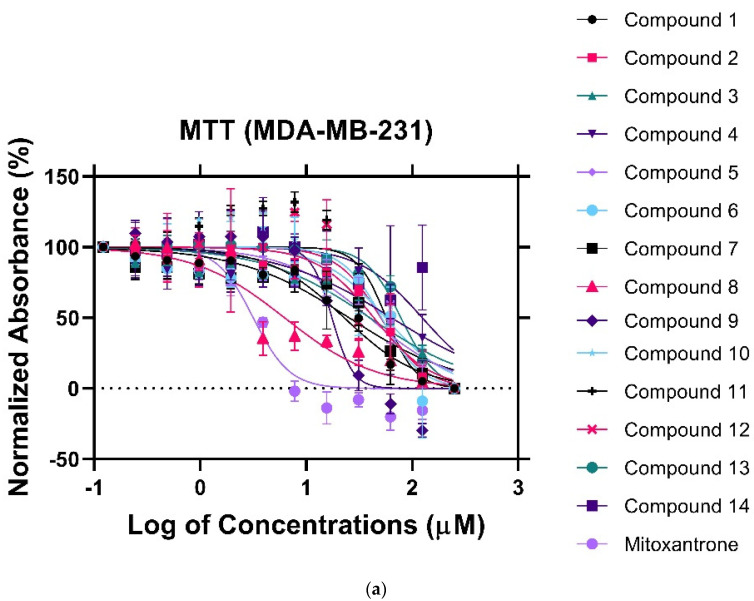
MTT assay of the fourteen (14) synthetic compounds against (**a**) MDA−MB−321 cancer cell lines, (**b**) MCF10A cell lines, (**c**) HCT8 cancer cell lines.

**Figure 4 molecules-27-02409-f004:**
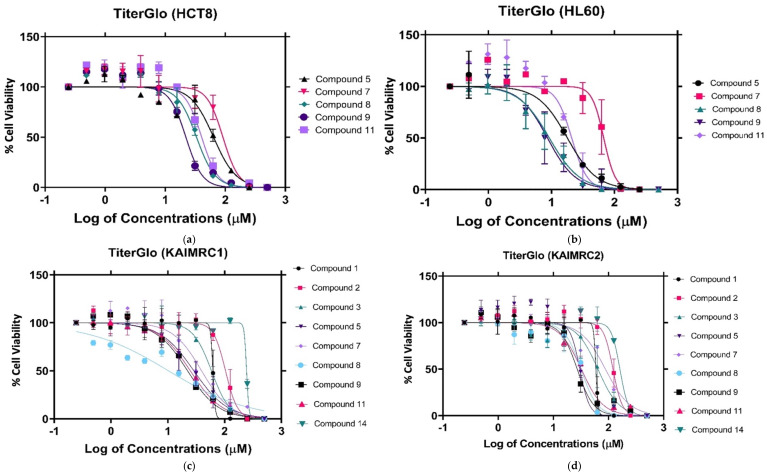
CellTiter-Glo assay against the following cell lines: (**a**) CellTiter-Glo assay against HCT8 cancer cell lines., (**b**) CellTiter-Glo assay against HL60 cancer cell lines., (**c**) CellTiter-Glo assay against KAIMRC1 cancer cell lines, (**d**) CellTiter-Glo assay against KAIMRC2 cancer cell lines, (**e**) CellTiter-Glo assay against MDA MB 231 cancer cell lines, and (**f**) CellTiter-Glo assay of mitoxantrone positive control.

**Figure 5 molecules-27-02409-f005:**
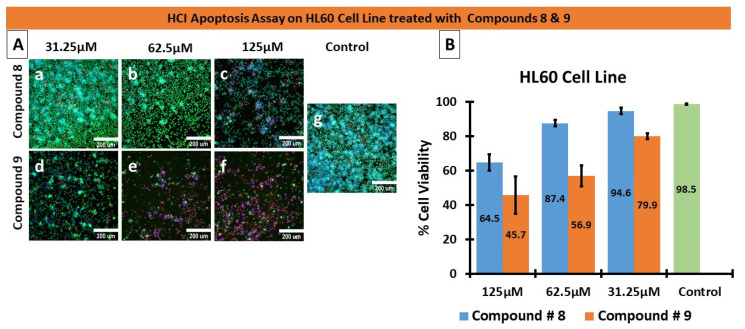
HCI-based apoptosis assay. (**A**) HL60 cells were treated with three concentrations (31.25 µM, 62.5 µM, and 125 µM) of compounds # 8 (**a**–**c**), # 9 (**d**–**f**), and PBS for the negative control (**g**) for 48 h. The nucleus (blue) was stained with HOECHST 33342, the cytoplasm (Green) was stained with Calcein AM, and dead cells (Red) were stained with Propidium Iodide (PI). Compound # 9 was more potent than compound 8, indicated by the presence of dead cells (red) and diminished Calcein AM (Green) staining. (**B**) HCI Cell Health Digital Analysis MetaXpress software generated the graph quantifying the cell viability, seen in B, as a percentage of HL60 cells treated with the compounds. At low concentrations of 31.25 µM, compound 9 displayed approximately 20% cytotoxic efficacy and approximately 54% cytotoxicity at 62.5 µM concentrations.

**Figure 6 molecules-27-02409-f006:**
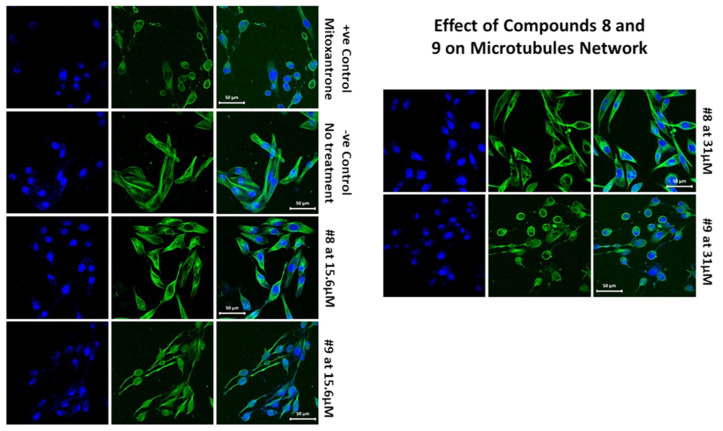
Effects of compounds 8 and 9 on the microtubular network.

**Figure 7 molecules-27-02409-f007:**
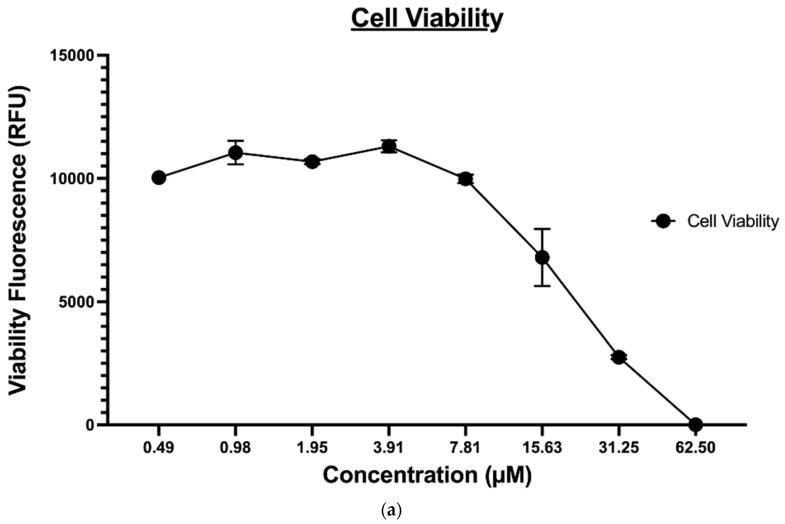
(**a**) The effect of compound 9 on MDA-MB-231 cell viability, (**b**) The effect of compound 9 on MDA-MB-231 cell apoptosis, and (**c**) The effect of compound 9 on MDA-MB-231 cytotoxicity.

**Figure 8 molecules-27-02409-f008:**
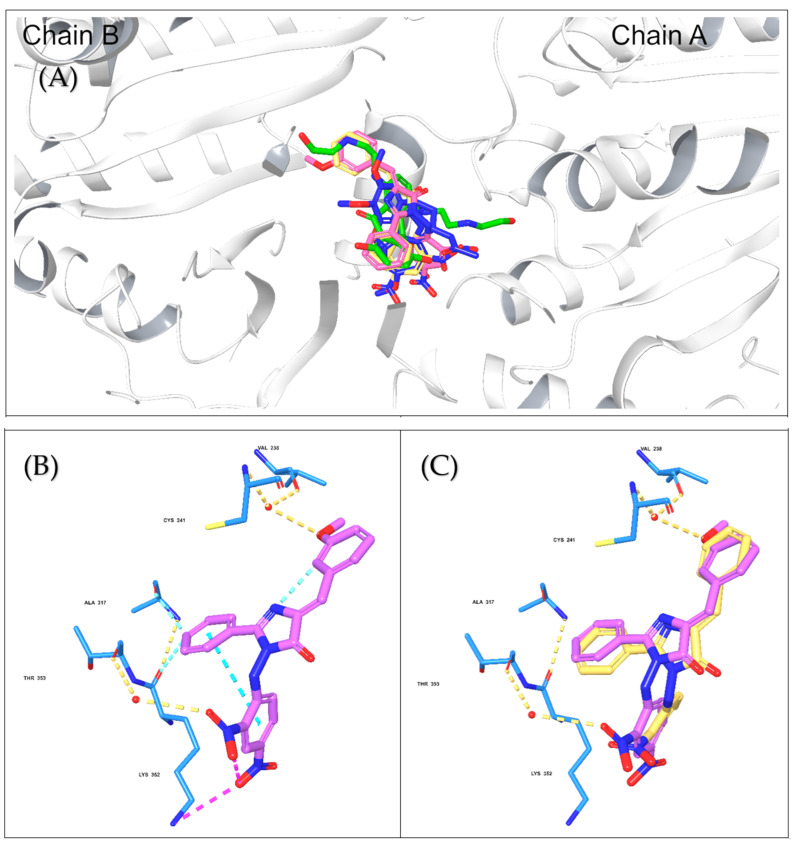
The molecular docking interactions for compounds 8 and 9 with tubulin crystal structure using Glide Maestro software. (**A**) Colchicine (blue), mitoxantrone (green), and compounds 8 (faded orange) and 9 (faded plum) superimposed onto the β-tubulin chain to highlight the similar binding site. (**B**) The molecular interactions of compound 9 (faded plum) with tubulin amino acid residues via H-bonds and Pi–Pi interactions. (**C**) Overlay of compounds 8 (faded orange) and 9 at the tubulin-binding site.

**Figure 9 molecules-27-02409-f009:**
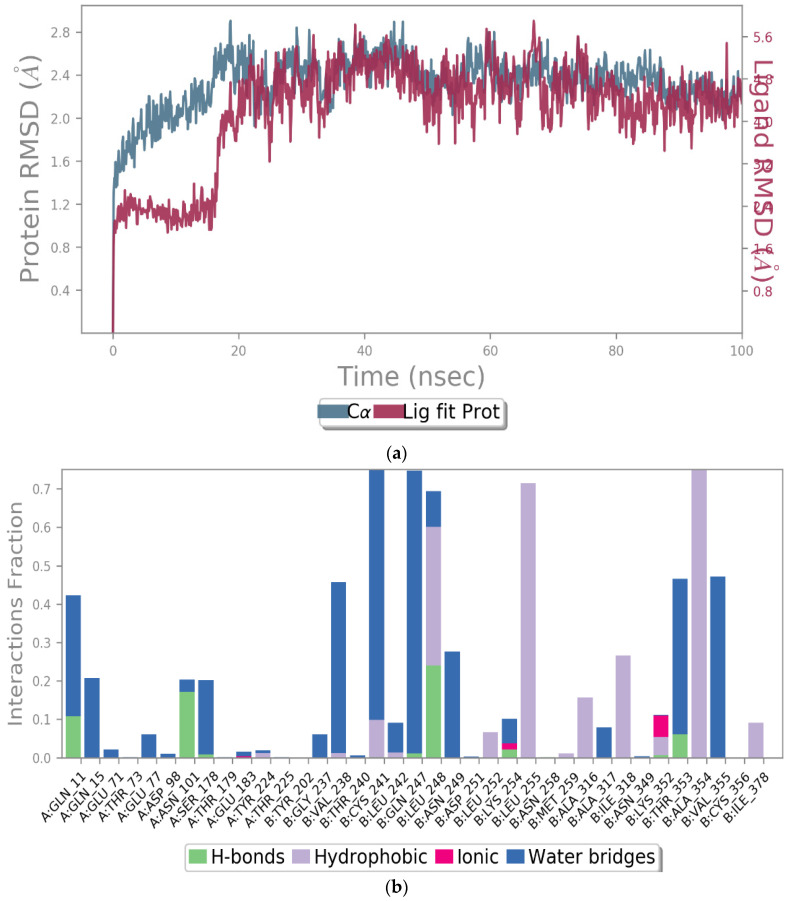
The simulation interaction diagram for compound 9 with the tubulin crystal structure over 100 ns. (**a**) the RMSD values for compound 9 with tubulin for a timescale of 100 ns, (**b**) the amino acid interactions of compound 9 with tubulin for a timescale of 100 ns, and (**c**) the ligand-protein contacts between compound 9 and tubulin crystal structure for a timescale of 100 ns.

**Table 1 molecules-27-02409-t001:** The IC_50_ (μM) of nitrogen-based derivatives against cancer cell lines using MTT assay.

Compound Names	Breast Cancer	Non-Malignant Breast Epithelial Cells	Colorectal Cancer
	(MDA-MB-231)	(MCF-10A)	(HCT8)
**1**	22.88	N/A	51.15
**2**	43.55	65.5	72.22
**3**	39.95	NA	N/A
**4**	69.44	NA	N/A
**5**	39.56	34.20	39.06
**6**	50.65	50.38	70.83
**7**	22.96	62.99	73.55
**8**	4.759	30.66	29.53
**9**	17.02	9.516	13.20
**10**	56.02	N/A	N/A
**11**	57.15	N/A	N/A
**12**	50.75	N/A	N/A
**13**	78.46	N/A	N/A
**14**	138.7	N/A	N/A
Mitoxantrone	3.171	2.898	0.7113

N/A: not applicable.

**Table 2 molecules-27-02409-t002:** The IC_50_ (μM) of nitrogen-based derivatives against five cancer cell lines obtained using the CellTiter-Glo assay.

Compound.	Leukemia	Breast Cancer			Colorectal Cancer
	HL60	MDA-MB-231	KAIMRC1	KAIMRC2	HCT8
**1**	N/A	N/A	62.21	59.21	N/A
**2**	N/A	N/A	103.6	120.6	121.3
**3**	N/A	N/A	58.56	64.32	N/A
**4**	N/A	N/A	N/A	N/A	N/A
**5**	17.86	52.13	28.64	30.30	63.70
**6**	NA	N/A	N/A	N/A	N/A
**7**	67.43	60.91	39.18	87.93	91.22
**8**	9.237	18.97	10.20	31.64	33.11
**9**	8.632	39.19	22.18	27.11	22.05
**10**	N/A	N/A	N/A	N/A	N/A
**11**	20.92	45.42	24.81	32.16	40.35
**12**	N/A	N/A	N/A	N/A	N/A
**13**	N/A	N/A	N/A	N/A	N/A
**14**	N/A	242.7	243.4	161.2	N/A
Mitoxantrone	0.1252	1.936	1.713	0.8008	5.618

N/A: not applicable.

**Table 3 molecules-27-02409-t003:** Anticancer activity and molecular target predictions obtained using the PASS Online and SEA Search webservers.

Compound	PASS Online		SEA Search Server	
	Anticancer		Microtubule-Associated Protein Tau	
	Pa	Pi	*p* Value	MaxTC
**1**	Inactive	Inactive	Inactive	
**2**	Inactive	Inactive	9.42 × 10^−7^	0.37
**3**	0.258	0.180	Inactive	
**4**	0.242	0.192	1.864 × 10^−38^	0.43
**5**	0.224	0.208	Inactive	
**6**	0.364	0.119	2.801 × 10^−33^	0.44
**7**	Inactive	Inactive	Inactive	
**8**	0.413	0.099	1.544 × 10^−6^	0.32
**9**	0.434	0.092	1.137 × 10^−6^	0.34
**10**	0.411	0.100	Inactive	
**11**	0.498	0.072	3.343 × 10^−26^	0.36
**12**	Inactive	Inactive	Inactive	
**13**	0.495	0.073	7.011 × 10^−10^	0.34
**14**	0.493	0.074	Inactive	

**Table 4 molecules-27-02409-t004:** The Glide docking scores and prime MM-GBSA energy properties.

Compound Name	Docking Scores (Kcal/mol)	MMGBSA dG Bind (Kcal/mol)
Colchicine	−10.10	−88.61
Mitoxantrone	−10.42	−42.31
Compound 8	−7.56	−22.47
Compound 9	−7.40	−42.35

**Table 5 molecules-27-02409-t005:** Predictions of the ADME properties for the 14 N-heterocycle derivatives using SwissADME and QikProp computational tools.

Compound	GI Absorption		BBB Penetration		P-gp Substrate		Log Po/w		Log S	
	SwissADME	Qikprop (% Absorption)	SwissADME	Qikprop	SwissADME	Qikprop	SwissADME	Qikprop	SwissADME	Qikprop
**1**	High	100	Yes	0	No	N/A	2.54	2.87	−5.44	−3.529
**2**	N/A	100	N/A	0	N/A	N/A	3.10	3.586	N/A	−4.385
**3**	High	100	Yes	0	No	N/A	2.55	2.869	−5.56	−3.383
**4**	High	100	Yes	0	No	N/A	2.61	3.249	−5.54	−4.305
**5**	High	92.21	Yes	−1	No	N/A	2.25	2.81	−4.86	−3.657
**6**	High	91.67	Yes	−1	No	N/A	2.26	2.459	−4.98	−3.721
**7**	High	100	Yes	0	No	N/A	2.13	2.105	−4.66	−2.525
**8**	Low	80.14	No	−2	No	N/A	3.10	3.049	−6.26	−5.222
**9**	Low	67.92	No	−2	No	N/A	3.11	3.174	−6.35	−5.164
**10**	Low	52.27	No	−2	No	N/A	2.80	2.607	−5.67	−5.677
**11**	Low	55.65	No	−2	No	N/A	2.81	2.520	−5.76	−4.877
**12**	Low	62.87	No	−2	No	N/A	2.69	2.382	−5.48	−4.231
**13**	High	95.79	Yes	−1	No	N/A	1.33	2.452	−4.79	−3.936
**14**	High	96.06	Yes	0	No	N/A	1.32	2.501	−4.68	−3.766

N/A: not applicable (the software did not provide any prediction).

**Table 6 molecules-27-02409-t006:** The CYP enzyme-inhibition profile for the 14 N-heterocycle Derivatives using the SWISSADME webserver.

Compound	CYP 1A2	CYP 2C19	CYP 2C9	CYP 2D6	CYP 3A4
**1**	Yes	Yes	Yes	No	No
**2**	N/A	N/A	N/A	N/A	N/A
**3**	Yes	Yes	Yes	No	No
**4**	Yes	Yes	Yes	No	No
**5**	Yes	No	No	No	No
**6**	Yes	Yes	Yes	No	Yes
**7**	Yes	Yes	No	No	No
**8**	No	Yes	Yes	No	No
**9**	No	Yes	Yes	No	Yes
**10**	No	Yes	Yes	No	No
**11**	No	Yes	Yes	No	Yes
**12**	No	Yes	Yes	No	Yes
**13**	Yes	No	No	No	No
**14**	Yes	No	No	No	No

N/A: not applicable.

**Table 7 molecules-27-02409-t007:** In silico toxicity analysis of the 14 synthesized compounds using the ProTox-II webserver together with their color key.

Compound	Oral Toxicity		Prediction of Active Organ Toxicity and Toxicity Endpoints	Probability
	Predicted LD_50_ (mg/kg)	Predicted Toxicity Class		
**1**	1190	4	Hepatotoxicity	0.69
			Immunotoxicity	0.96
			Aromatase	1.0
			Estrogen Receptor Alpha (ER)	0.99
			Estrogen Receptor Ligand-Binding Domain (ER-LBD)	1.0
**2**	3700	5	Hepatotoxicity	0.59
			Carcinogenicity	0.56
**3**	1400	4	Hepatotoxicity	0.55
			Carcinogenicity	0.57
**4**	5000	5	Inactive	-
**5**	3700	5	Hepatotoxicity	0.61
			Carcinogenicity	0.58
**6**	978	4	Hepatotoxicity	0.56
			Carcinogenicity	0.57
**7**	3500	5	Hepatotoxicity	0.59
			Carcinogenicity	0.54
**8**	600	4	Hepatotoxicity	0.63
			Carcinogenicity	0.68
			Mutagenicity	0.82
**9**	600	4	Hepatotoxicity	0.62
			Carcinogenicity	0.55
			Immunotoxicity	0.62
			Mutagenicity	0.78
			Mitochondrial Membrane Potential (MMP)	0.57
**10**	600	4	Hepatotoxicity	0.63
			Carcinogenicity	0.55
			Mutagenicity	0.80
			Mitochondrial Membrane Potential (MMP)	0.55
**11**	600	4	Hepatotoxicity	0.64
			Carcinogenicity	0.54
			Immunotoxicity	0.70
			Mutagenicity	0.77
			Mitochondrial Membrane Potential (MMP)	0.55
**12**	600	4	Hepatotoxicity	0.64
			Carcinogenicity	0.73
			Mutagenicity	0.83
			Mitochondrial Membrane Potential (MMP)	0.57
**13**	800	4	Hepatotoxicity	0.66
			Carcinogenicity	0.58
			Mutagenicity	0.51
**14**	800	4	Hepatotoxicity	0.67
			Carcinogenicity	0.71
Color key				
Class 4:	Harmful if swallowed (300 < LD_50_ ≤ 2000)			
Class 5:	It may be harmful if swallowed (2000 < LD_50_ ≤ 5000)			

## Data Availability

The data presented in this study are available on request from the corresponding author.

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
