# Peer review of "Distinct Mechanisms of Cytotoxicity in Novel Nitrogenous Heterocycles: Future Directions for a New Anti-Cancer Agent"

_molecules, 2022, doi:10.3390/molecules27082409_

Round 1
Reviewer 1 Report
Generally it is interesting paper on anticancer activity of new heterocyclic compounds which are derivatives of imidazole and oxazolone. The influence of 14 compounds on cells of different cancer cel lines was studied and cytotxic effect and inhibition of proliferation was observed. Aditionally an effect of the compounds on apoptosis and microtublule function was also studied. Authors used also computational simulation to predict anticancer activity of 14 compounds and to show molecular docking of the most active compounds with tubulin. Authors moreover utilized the software SWISS/ADME and QikProp to predict pharmacological properties of the newly synthesized compounds.
However according to the referee this paper needs some additional information and correction as follow:.
- There are no information about studied here newly synthetized compounds. Are these compounds a gift or are commersially available? What was their purity and chemical characteristics?
- Why MTT assay is compare with CellTiter-Glo assay not for the same set of cancer cell lines?
- English needs some corrections by native speaker.
- There are lacking bracket at the numbers of the cited papers in several places of the manuscript.
- In this paper is lacking the chapter Discussion
Author Response
Dear Respected Editor,
Thank you for taking the time to read and comment on our paper. We appreciate the time and work you and the other reviewers put into providing us with constructive feedback on the manuscript. We were able to properly incorporate all the necessary adjustments in order to reflect the reviewers' suggestions. We've also marked the changes that have been made to the document.
The following is a detailed response to the reviewers' questions and comments.

Reviewer 2 Report
Comments: - The manuscript deals with the study of Cytotoxicity of Nitrogenous Heterocycles compounds as a novel anti-cancer agent. The manuscript is well collated and written. It could be an interesting piece of information for the journal. The overall quality of the work is excellent, however, manuscript requires careful formatting. Many irregularities in formatting has been observed and the manuscript can be accepted only after doing those improvements.
Major comments:
- Author should use a positive control for the docking studies as well. May be Mitoxantrone could have been considered as positive control in docking studies too.
- Why did the author not calculate free binding energy, it would certainly give an interesting result. molecular mechanics generalized Born surface area(MM/ GBSA) or linear response approximation (LRA) can be used.
Minor comments:
- The affiliation identifiers can be changed to “Rizwan Ali2, 1” to “Rizwan Ali1,2”
- Please check affiliations of all authors:- Change “Sarah Hwaizi2, Shatha Algheribe2, Zeyad Alehaideb2 and Imadul Isalm2.” To “Sarah Hwaizi2, Shatha Algheribe2, Zeyad Alehaideb2 and Imadul Isalm2”
- No author has been designated with affiliation identifier 3 Please check
- Please check the reference style in the text throughout the manuscript. Some references are in brackets while some are not. Please make it consistent.
- Kindly enlarge the graph of figure 3A, B, C for better readability if possible or provide higher resolution images
- In some figure, sub sections are labelled as “a, b, c..” while in some they are labelled as “A, B, C..”. Please make it consistent.
- Please keep the headings as pert the template provided by Molecules, MDPI. For example : 1. Introduction and Background” can be changed to “1. Introduction” for better representation
- Line 55 : Please check the sentence “annual number of new cases, possibly reaching 27.5 million annually by 20407.”
- Line 65: Please check the sentence “with a greater affinity 12. In addition, imidazoles have some structural similarities to”. Is “12” a reference?
- gram positive/negative must be written as Gram-positive/ negative
- Line 110-111 rephrase the sentence
- Line 145: 18.97 cannot be left without the unit
- Line 150: article in the middle of the sentence must start with lower case ( An/ an AKT)
- Mention the whole word before using abbreviation for the first time (AKT)
- references 33, 37-40, 46 is not as per the format of the journal
- Word ‘Table 5’ should not be in italic
- Please make the units consistent throughout the manuscript. Several places it is “hours” and also “h”; please make it uniform
- Please check the manuscript for units and symbols, many inconsistencies have been observed
- Please use a standard format for writing amino acids ( for example CYS241 should be written as Cys 241)
Author Response

(The authors gave the same response as above.)

Reviewer 3 Report
The article discusses the specific mechanism of cytotoxicity in nitrogen-containing heterocycles and future guidelines for the development of novel antineoplastic agents. Experiments were nicely conducted. However, before proceeding for acceptance, there are a few concerns that need to be addressed as mentioned below: -
- Lots of typos were found in the manuscript, including materials and methods. It seems the authors didn’t proofread before submitting it to the journal.
- Line 90: - Compounds 8 and 9 are improperly structured, rectify the structure.
- For cell-titer Glo assay, the x-axis should be % viability/ relative viability. Please change that. The authors have used to describe Cell-titer Glo assay, the methods need to rectify. Please provide the correlation between μg/ml and μ
- Line 149-151: - Provide experimental evidence for the claim “Both KAIMRC cell lines displayed differing proteomic phosphorylation statuses compared to MDA-MB-231 and featured constitutive activation of An Akt enzyme is a type of serine/threonine-protein kinase (AKT) in a ligand-independent manner.
- References must be provided in manuscripts wherever required. Few examples: Line 66, Line 152-154, Line 166-168, Line 193-194…
- Provide the method for safety Profile Analysis- CYP P450 Enzyme Inhibition.
- In lines 339-354, the authors mentioned that compounds 8 and 9 are harmful and unsafe orally and cause organ and endpoint toxicity. The statements made in those mentioned line is indecisive and the main purpose of compounds 8 and 9 are abrogated. Justify with strong evidence for the claim.
- In conclusion, the authors claim that compound 9 exhibited an acceptable pharmacokinetic profile, please provide a reason for that statement or change the conclusion accordingly. There is an ample amount of lead optimization to be done to defend the anti-cancer properties of compound 9.
Author Response

(The authors gave the same response as above.)

Round 2
Reviewer 3 Report
The authors have improved the manuscript. I have no major concerns.